# Study on the Mechanical and Thermal Properties and Shape Memory Behaviors of Blends of Bio-Based Polybenzoxazine and Polycaprolactone with Different Molecular Weights

**DOI:** 10.3390/polym16233391

**Published:** 2024-11-30

**Authors:** Sunan Tiptipakorn, Naritsara Chaipakdee, Sarawut Rimdusit, Kasinee Hemvichian, Pattra Lertsarawut

**Affiliations:** 1Department of Physical and Material Sciences, Faculty of Liberal Arts and Science, Kasetsart University, Nakhon Pathom 73140, Thailand; naritsara.chai@ku.th; 2Center of Excellence in Polymeric Materials for Medical Practice Devices, Department of Chemical Engineering, Faculty of Engineering, Chulalongkorn University, Bangkok 10330, Thailand; sarawut.r@chula.ac.th; 3Thailand Institute of Nuclear Technology (Public Organization), Ongkarak District, Nakornnayok 26120, Thailand; kasinee@tint.or.th (K.H.); pattra@tint.or.th (P.L.)

**Keywords:** shape memory systems, bio-based polybenzoxazine, polycaprolactone, different molecular weights

## Abstract

In this research, blends of bio-based polybenzoxazine (V-fa) and polycaprolactone (PCL) with different molecular weights (M_n_) (14,000, 45,000, and 80,000 Da) were prepared with varying PCL content from 10 to 95 wt%. The spectra measured using Fourier Transform Infrared Spectroscopy (FTIR) may indicate the presence of hydrogen bonding between two polymeric components. The thermograms obtained using a Differential Scanning Calorimeter (DSC) and dynamic mechanical analyzer (DMA) exhibited a shift in glass transition temperature (T_g_), which indicated partial miscibility between V-fa and PCL. The thermograms obtained using a thermogravimetric analyzer (TGA) revealed that the addition of PCL led to an increase in the maximum decomposition temperature (T_dmax_). The tensile strength and modulus decreased with an increase in PCL, thus indicating a decrease in brittleness. Interestingly, only the samples with an M_n_ of 80,000 Da were bendable. The blends with 80 wt% PCL were revealed to have shape memory behaviors with a shape fixity of approximately 81%. The shape recovery ratio of the blends with 95 wt% PCL was approximately 78%.

## 1. Introduction

For decades, polybenzoxazines (PBZs) have been known as a type of phenolic resin, which possess many advantages, such as their lack of need for a catalyst during ring-opening polymerization. In addition, no by-products are generated upon curing, leading to the absence of voids in products. These materials can be synthesized through the reaction of amine compounds and formaldehyde in the presence of phenolic components, thus leading to the creation of polybenzoxazine products after fabrication with near-zero volumetric shrinkage and great thermal stability, good chemical resistance and mechanical properties, and low water absorption [1,2,3,4,5,6]. These novel thermosets can be categorized into two types, i.e., petroleum-based and bio-based ones. Bio-based polybenzoxazine is a type of phenolic resin derived from renewable resources, providing a sustainable alternative to petroleum-based benzoxazine resins. The production approach for bio-based materials, which are derived wholly or partly from natural resources [7], not only decreases dependence on fossil fuels but also reduces the environmental impact during production [8,9,10]. Bio-based polybenzoxazines can be applied in various fields, such as the food packaging, textile, paint, automotive, aerospace, and electronic industries, in which high-performance materials are required [11,12]. However, to extend the applications of these thermosets, their toughness must be enhanced. Typically, the brittleness of PBZ can be reduced using two major approaches, i.e., (i) molecular design and (ii) bending or alloying it with other components [13,14]. The latter technique is versatile and relatively uncomplicated for improving the thermal and mechanical properties of polybenzoxazine [14]. Recently, the toughness of PBZ has been enhanced by blending and alloying it with other polymeric components, such as epoxy, dianhydride, polyimide, and polyurethane [14].

Polycaprolactone (PCL), a biodegradable polyester, is derived from the polymerization of ε-caprolactone. Due to its flexibility, biocompatibility, and low melting point, it is possibly a good candidate for blending with PBZ. Ishida and Lee [3,4] prepared blending systems between petroleum-based polybenzoxazine (PBA-a type) and poly(ε-caprolactone). They reported that ring-opening polymerization and the subsequent reaction were majorly facilitated by the incorporation of PCL. The formation of hydrogen bonds between the hydroxyl groups of PBZ and carbonyl groups of PCL was observed via FTIR. The crosslink densities of the blends were higher than those of neat PBZ [4]. Huang and Yang [5] studied the thermal behavior of petroleum-based polybenzoxazine (B-m type) and PCL. They also reported hydrogen bonding between two polymeric components. With increasing PCL content from 0 to 40 wt%, the crystallinities and glass transition temperatures of the blends increased [5]. Tiptipakorn et al. studied the thermal and mechanical properties of PBA-a and PCL with various molecular weights [15]. They reported on the synergistic behaviors of the glass transition of the blends. The blends with higher PCL molecular weight had higher glass transition temperatures among the blends. Although the blends between PCL with different molecular weights and petroleum-based polybenzoxazine were studied in a previous study [15], there was no study of the effects of PCL molecular weight on mechanical and thermal properties, including the shape memory behaviors of blends derived from bio-based polybenzoxazine and PCL. Blending between bio-based PBZ and PCL could merge the benefits of both polymeric materials, creating environmentally friendly blends with improved properties.

Shape memory polymers (SMPs) are a broad class of smart polymeric materials that possess the ability to recover their initial permanent shape after being deformed upon exposure to a suitable external trigger [16], such as thermal change [17,18,19], light [20,21,22,23,24], pH change [25], or a magnetic field [26,27]. SMPs can be used in various applications, such as controllable remote actuation, biomedical sensors and devices, automotive components, smart textiles, and robotics, including deployable parts for aircraft and spacecraft [28]. Moreover, SMPs have been utilized instead of shape memory metal alloys due to their many benefits, such as their ease of fabrication, light weight, low cost, and high flexibility [29]. Typically, SMPs are composed of hard and soft segments. A hard segment cooperates with a stable network, while a soft segment is related to a reversible switching transition, such as melting or crystallization transition (T_m_) or a glassy/rubbery transition (T_g_) [30]. SMPs based on PBZ have gained a lot of attention due to their high-performance properties and chemical network structure, which is determined as the stationary phase (hard segment) in SMPs. Gu and Jana studied the effect of benzoxazine content on the hard moiety of poly(benzoxazine-urethane) [31]. They reported that an increase in benzoxazine (BA-a type) content could lead to an increase in hard moiety in blends. When increasing BA-a content, the glass transition temperature of the blends increased, and the number of hydrogen bonds decreased. The shape fixity of SMPs decreased with increasing polybenzoxazine, i.e., from 97.2 to 92.8%. When increasing BA-a content, the recovery ratio increased, from approximately 60 to 93% [31]. Recently, Luo et al. prepared polybenzoxazines (P-pea and P-fa types) blended with modified **ε**-polycaprolactone. They reported on triple-shape memory behaviors with a high char yield of 45% at 800 °C [32]. Prasomsin et al. prepared bio-based benzoxazine (V-fa type)/epoxy composites reinforced with carbon nanotubes. They reported that the composites could present NIR laser-stimulated shape memory behaviors [8].

In recent years, benzoxazine monomers have been successfully synthesized from renewable resources such as vanillin, eugenol, furfurylamine, guaiacol, and stearylamine [8]. In this study, vanillin and furfurylamine were the starting chemicals for the synthesis of bio-based benzoxazine. Bio-based benzoxazine (V-fa type) presented a low polymerization temperature of approximately 179 °C. It possesses structures with high crosslink density, leading to a high glass transition temperature. Moreover, bio-based benzoxazine (V-fa type) provides a higher char yield than petroleum-based benzoxazine [33]. PCL with different molecular weights has been proposed for blending with bio-based benzoxazine (V-fa type) to extend the utilization range of thermosetting materials with comparatively simple paths and no structure modification. The effects of PCL content and the molecular weight of PCL on the thermal and physical properties of blending systems were investigated as well.

## 2. Experimental Section

### 2.1. Materials and Methods

Bio-based benzoxazine resin (V-fa type) was derived from vanillin, furfurylamine, and paraformaldehyde. Vanillin (99% purity) and furfurylamine (99% purity) were pur-chased from TCI America (Portland, OR, USA). Formaldehyde (AR grade) and Tetrahy-drofuran (Ph Eur grade) were bought from Merck Co., Ltd. (Darmstadt, Germany). Poly-caprolactone with molecular weights (M_n_) of 14,000 Da (denoted as 14 k), 45,000 Da (de-noted as 45 k), and 80,000 Da (denoted as 80 k) were supplied from Sigma-Aldrich Pte. Ltd. (Singapore).

### 2.2. Preparation of Bio-Based Benzoxazine Monomers and PCL/V-fa Blends

To synthesize a bio-based benzoxazine monomer (V-fa), vanillin, furfurylamine, and paraformaldehyde (at a molar ratio of 1:1:2) were homogeneously mixed according to the solvent-free synthesis method [1,33]. All mixed components were heated at 105 °C for 60 min. After cooling down the mixtures, the resulting yellow monomer was obtained in solid form. The monomer was then dissolved in THF and blended with PCL at weight ratios ranging from 10 to 95%. The blends were poured into Teflon molds and heated in an oven at 60 °C (18 h), 110, 120, 130, 140 °C (20 min for each step), 150 °C (1 h), 160 °C (1 h), 170 °C (2 h), and 180 °C (2 h). The obtained sheet was left at room temperature before being removed from the mold.

### 2.3. Characterizations

The structural characterizations for the V-fa monomer and polymer poly(V-fa) were conducted using a ^13^C solid-state Fourier Transform Nuclear Magnetic Resonance Spectrometer (NMR, Bruker, model Avance III HD/Ascend 400 WB, Mannheim, Germany) at 400 MHz. The functional groups of the blends were investigated using a Fourier Transform Infrared (FTIR) Spectrometer (Bruker, model Tensor 27, Mannheim, Germany) with a ZnSe crystal in ATR mode at a scan number of 32 and resolution of 4 cm^−1^.

To determine dynamic mechanical properties, a dynamic mechanical analyzer (DMA, Mettler Toledo, model DMA 1 STARe system, Greifensee, Switzerland) was used. The dimensions of the sample for tensile mode were kept at approximately 25 × 5 × 1 mm^3^. The tested sample was heated at 2 °C·min^−1^ from −100 to 40 °C with a deformation frequency of 1 Hz under a nitrogen atmosphere. The peak on the tan delta curve of DMA thermograms could present the glass transition temperature (T_g_). The melting temperature and heat of melting of the sample were determined using a Differential Scanning Calorimeter (DSC, Netzsch, model NETZSCH DSC 204 F1 Phoenix, Selb, Germany). Approximately 5 mg of the tested sample was heated at 20 °C/min from −100 to 250 °C with a nitrogen flow of 60 mL/min. A thermogravimetric analyzer (TGA, Mettler Toledo, model TGA1 STARe System, Greifensee, Switzerland) was used to determine the thermal stability of blending materials. The heating program at 20 °C/min from 25 to 800 °C was conducted for approximately 10 mg of the sample. The decomposition temperature at the maximum decomposition rate and char yield at 800 °C were reported. The flexural properties of the samples (with dimensions of 10 × 60 × 2 mm^3^) were tested using a Universal Testing Machine (UTM, Hounsfield, model H10 KM, Redhill, UK) according to ASTM D790 [34]. The morphology of the fracture surface was investigated using a Scanning Electron Microscope (SEM, JEOL, model JSM-IT500HR, JEOL Ltd., Akishima, Tokyo, Japan).

Shape memory behaviors were presented in three parameters, i.e., shape fixity ratio (R_f_), shape recovery ratio (R_r_), and recovery time. Firstly, samples in the form of a rectangular sheet with dimensions of 60 × 10 × 1 mm^3^ were heated to a temperature of T_g_ + 20 °C and bent to be L-shaped (temporary shape) with an angle of around 90°. The samples were then left at room temperature; after unloading, the angle was measured as *θ*_1_. The shape fixity ratio was calculated using Equation (1). For the calculation of the shape recovery ratio, each sample in a temporary shape was placed in an oven at a temperature of T_g_ + 20 °C. The shape recovery ratio was calculated using Equation (2), where *θ*_2_ represents the angle after the recovery step was completed [35].
(1)$$Shape\;fixity\;ratio\;\left(R_{f}\right)=\frac{\theta_{1}}{90^{o}}\times 100$$
(2)$$Shape\;recovery\;ratio\;\left(R_{r}\right)=\frac{\left(\theta_{1}-\theta_{2}\right)}{\theta_{1}}\times 100$$

## 3. Results and Discussion

### 3.1. Investigation of Chemical Structures

The solid-state ^13^C NMR spectra and the chemical structure of the V-fa monomer and poly(V-fa) are presented in Figure 1.

Based on the data obtained from solid-state ^13^C NMR spectroscopy, it can be concluded that carbon chemical shifts in the oxazine ring in the V-fa monomer can be observed at the peaks of CH_2_N at 49.48 ppm (1) and CH_2_O at 80.30 ppm (2). Aromatic structures can be observed at the peaks in the range from 118.62 to 148.57 ppm for monomers and 128.13 to 151.46 ppm for poly(V-fa) [36]. The peaks at approximately 190 ppm (3) and 54.99 ppm (4) can indicate the functional groups of R-CO-H and the methyl group, respectively. Furan rings can be observed from the peaks of 109.64–111.41 ppm (5) and 107.51 ppm (6). The peak at 45.80 ppm (7) indicates the CH_2_ group connecting with the amine group [37].

The chemical structures of the V-fa monomer and poly(V-fa), including the chemical interactions between the V-fa and PCL components, were investigated using the ATR-FTIR technique. In the spectrum of pure PCL, the broad band at the wavenumber in the range of 2861–2942 cm^−1^ typically corresponds to the stretching of C-H bonds in aliphatic polymeric chains. The signal at 1165 cm^−1^ corresponds to the stretching vibration of C-O bonds. For the spectra of the synthesized V-fa monomer and poly(V-fa), the signal at the wavenumbers of 3441, 1682, and 1674 cm^−1^ indicates the presence of carbonyl (C=O) groups. The signal of 1300 cm^−1^ in the poly(V-fa) spectrum indicates C-N stretching. The ring-opening polymerization of V-fa monomers was evidenced by the disappearance of the peak of the oxazine ring at 1248 and 910 cm^−1^, which are assigned to the asymmetric and symmetric stretching modes of C-O-C, respectively [8,36]. The infrared spectra of the blends with different content of 45 k PCL are presented in Figure 2. It could be noticed that the peak which indicated a carbonyl group shifted when the PCL content was varied; i.e., 1724 cm^−1^ (for 90% PCL) shifted to 1721 cm^−1^ (for 50% PCL). The chemical interaction proposed as hydrogen bonding is presented in Figure 3. This phenomenon corresponds to the blending system between petroleum-based polybenzoxazine (PBA-a) and polycaprolactone (PCL) in previous studies [3,4,8,15].

### 3.2. Determination of Glass Transition Temperature and Melting Temperature

The glass transition temperature (T_g_) values of the V-fa/PCL blends are related to the switching temperature of the shape memory materials. In this research, T_g_ could be determined from the peak of loss tangent in DMA thermograms (Figure 4). It could be noticed that a decrease in V-fa could increase T_g_, i.e., −50, −43, and −30 °C for 95, 90, and 80%, respectively. Also, the storage modulus at 40 °C decreased with a decrease in PCL, i.e., 113.1, 91.7, and 66.1 MPa for 95, 90, and 80%, respectively. It could be implied that the increase in PCL content led to more toughness in the blends. This was attributed to the fact that adding rubbery PCL into rigid poly(V-fa) could decrease the brittleness of the blends. This phenomenon corresponds to the PBA-a/PCL blending systems with a PCL content from 0 to 40 wt% in our previous study [15].

The crosslink density of the blends can be calculated using Equation (3):(3)$$\nu =\frac{\;E^{\prime }}{3\mathrm{RT}}$$
where ν is crosslink density, R (gas constant) is 8.314 J·mol^−1^·K^−1^,  E ′ is the storage modulus (Pa), and T is temperature (K). The value of crosslink density increased with a decrease in PCL content, i.e., 57,302, 48,998, and 32,095 mol/m^3^ for 95, 90, and 80%, respectively. This phenomenon could be related to the low free volume of molecular motion with a high V-fa content, which results in a high glass transition temperature and crosslink density [5]. The trend in the relationship between crosslink density and PCL content was like that of the PBA-a/PCL blending system at 0–40 wt% [15].

From Figure 5a, it could be noticed that at the same PCL molecular weight, the addition of PCL content could increase the heat of melting of the blends, i.e., 147.7, 462.0, and 831.8 J/g for 50, 70, and 80 wt%, respectively. However, the melting temperatures of the blends had no significant change (at approximately 58–60 °C). From Figure 5b, it could be noticed that at the same PCL content, the increase in PCL molecular weight could decrease the heat of fusion of the blends, i.e., 524.1, 364.4, and 110.9 J/g for 14 k, 45 k, and 80 k PCL, respectively. The melting temperature slightly increased with an increase in PCL molecular weight. These results correspond to the fact that the melting enthalpy of pure 14 k, 45 k, and 80 k PCL was reported to be 76, 65, and 59 J/g. Moreover, it was reported that the melting temperatures of pure 14 k, 45 k, and 80 k PCL were 56, 57, and 63 °C, respectively. The percentages of crystallinity of pure 14 k, 45 k, and 80 k PCL were reported to be 78, 68, and 60%, respectively [14]. The glass transition temperature value was slightly increased with increasing PCL molecular weight, i.e., −54.1, −53.3, and −52.5 °C for 14 k, 45 k, and 80 k PCL, respectively. This could be due to the low free volume of molecular motion at a high molecular weight [5,15].

### 3.3. Mechanical Properties of V-fa/PCL Blends

As seen in Figure 6, the addition of PCL led to a decrease in the tensile modulus, i.e., 540.5 ± 52.9, 487.7 ± 8.0, 399.9 ± 27.3, and 370.5 ± 20.5 MPa at 80, 85, and 90 wt%, and a decrease in tensile strength of 31.6 ± 1.1, 28.9 ± 1.1, and 28.0 ± 2.2 MPa at 80, 85, and 90 wt%. The results of the tensile test corresponded to those of the flexural test, as shown in Table 1.

At the same PCL molecular weight, from the above table it can be noticed that the values of the flexural modulus and strength both decreased with the content of PCL. At the same PCL content, blends with a higher PCL molecular weight could also have a lower flexural modulus and higher flexural strength. In general, the molecular weight of polymers has substantial effects on their physical and mechanical characteristics. A polymer with a high molecular weight could have improved toughness and strength, in addition to enhancement of heat and chemical resistance. This is attributed to the fact that the entanglement of polymeric chains increased, leading to enhanced structural integrity. A polymer with a lower molecular weight tends to have lower viscosity with a greater ease of processability but lower mechanical strength. Also, it has been reported that polymers with low molecular weights possess higher crystallinity, leading to more brittleness and higher modulus values [38].

### 3.4. Determination of Thermal Stability of V-fa/PCL Blends

The thermal stability of the materials is presented in the parameters of the decomposition temperature and char yields. The residual weight of the V-fa/80 k PCL blends after heating from room temperature to 800 °C is presented in Figure 7. The decomposition temperature and char yield of the blends with different weights are shown in Figure 8a,b, respectively.

For the blends with different PCL molecular weights with the same PCL content (for both Figure 8a,b), it could be noticed that there was no significant difference in decomposition temperature at the maximum decomposition rate (T_dmax_) and char yield. An increase in PCL content could lead to an increase in T_dmax_, i.e., approximately 416, 419, 420, 422, and 424 °C for 10, 30, 50, 70, and 90 wt%, respectively. The char yield of the blends at 800 °C decreased with an increase in PCL content, i.e., 57, 38, 25, 14, and 6% for 10, 30, 50, 70, and 90 wt%, respectively. It could be noticed that PCL molecular weight has no effect on the thermal stability of the V-fa/PCL mixture. This phenomenon corresponds to the research of Ishida and Lee [4], who studied the thermal stability of a PBA-a/PCL mixture with a PCL content of 0–15 wt%. They reported that the decomposition temperature of the PBA-a/PCL mixture increased in the medium-temperature region ranging from 250 to 450 °C. Moreover, an increase in decomposition temperature and a decrease in char yield with an increase in PCL were also noted in the PBA-a/PCL mixture with a PCL content of 0–40 wt% [15].

### 3.5. Morphology of Fracture Surface of V-fa/PCL Blends

The fracture surface of the blends with different PCL molecular weights is presented in Figure 9. It could be noticed that the blends with higher PCL molecular weights had greater roughness. A smoother fracture surface (as seen in Figure 9a) indicated higher brittleness, while a rougher fracture surface (as seen in Figure 9c) indicated a higher flexibility of the blends. Pure PCL with a lower molecular weight was reported to provide a higher degree of crystallinity and a greater flexural modulus [15]. In general, strong intermolecular forces in crystalline portions led to rigidity and exposed the materials to brittle failure when subjected to stress [39]. The fracture surface appearances correspond to the values of the flexural modulus presented in the previous section.

### 3.6. Bendability of V-fa/PCL Blends

When bending the uniform rectangular thin sheets with the dimensions of a 600-micron thickness × 60 mm length × 10 mm width, the appearances of the samples at a PCL:V-fa weight ratio of 95:5 were as shown in Figure 10. It could be noted that the blending sample with a PCL molecular weight of 80,000 Daltons was bendable.

The V-fa/PCL blends with a PCL molecular weight of 80,000 Daltons were selected to be used for further investigations of shape memory behaviors. Furthermore, it could be observed that the blends with a PCL content lower than 80 wt% could not be fabricated in the form of films at a thickness less than 1 mm. Therefore, the tests of shape recovery behavior were conducted only for the blends at 80 k PCL with a PCL content in the range of 80–95 wt%.

### 3.7. Shape Memory Behaviors of V-fa/PCL Blends

The crucial parameters related to shape recovery behaviors are shape fixity, shape recovery, and recovery time. It can be seen from Figure 11 that shape fixity decreased, and the shape recovery ratio increased with PCL content, i.e., shape fixities of 80.5, 75.0, 72.2, and 63.9% and shape recoveries of 69, 72, 75, and 78% were reported at 80, 85, 90, and 95 wt% 80 k PCL, respectively.

A snapshot of the recovery process for the V-fa/80 k PCL blends is presented in Figure 12. The recovery time for the blends decreased with PCL content, i.e., 180, 175, 150, and 75 s at 80, 85, 90, and 95 wt%, respectively.

After the recovery process was repeated a varying number of times, the shape recovery ratio and recovery time were obtained, as presented in Table 2. It could be noticed that the shape recovery ratio decreased with an increase in cycle number, i.e., from 78% (for cycle 1) to 64% (for cycle 4) at 95 wt% PCL content. Meanwhile, the recovery time increased with an increasing cycle number, i.e., from 75 s (for cycle 1) to 170 s (for cycle 4) at 95 wt% PCL content.

The addition of V-fa into the blends could lead to an increase in shape fixity; this could be due to their aromatic structure, including their crosslinking structure, which could mean that V-fa acts as a hard segment in shape memory materials. Meanwhile, an increase in PCL content could lead to an increase in shape recovery due to the structure of PCL acting as a soft segment in shape memory materials. When the benzoxazine content increased, the shape recovery time increased because the molecular chain impeded increasing crosslink density [39,40].

## 4. Conclusions

Blends of V-fa bio-based polybenzoxazine and polycaprolactone of different molecular weights (M_n_) were successfully prepared. The FTIR spectra revealed the presence of hydrogen bonding between the two polymers. The glass transition temperature (T_g_) values of the V-fa/PCL blends increased when adding V-fa. The DSC thermograms exhibited partial miscibility between V-fa and PCL. The TGA thermograms showed that the maximum decomposition temperature (T_dmax_) increased and char yield decreased with PCL content. At the same PCL content, there was no significant difference in the T_m_, T_dmax_, or char yield of the blends at 800 °C with different molecular weights of PCL. At the same PCL molecular weight, the values of the flexural modulus and flexural strength decreased with PCL content. Moreover, at the same PCL content, the modulus decreased, and the strength increased with increasing molecular weights. It was observed that the thin sheet samples of the blend with an M_n_ of 80,000 Da were bendable. The shape fixity decreased, and the shape recovery ratio increased with the PCL content. For a PCL content of 80–95 wt%, the shape fixity was in the range of approximately 64–81%, while the shape recovery was in the range of approximately 69–78% and the recovery time was in the range of 75–180 s. In addition, the recovery time increased with an increasing number of recovery process cycles.

## Figures and Tables

**Figure 1 polymers-16-03391-f001:**
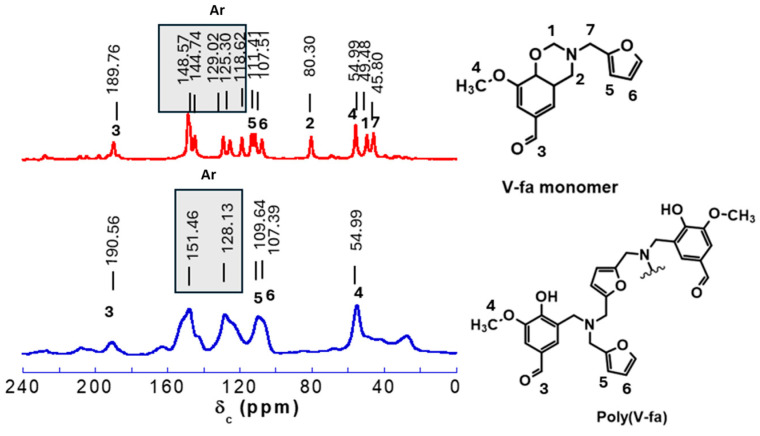
Solid-state ^13^C NMR spectra of V-fa monomer and poly(V-fa).

**Figure 2 polymers-16-03391-f002:**
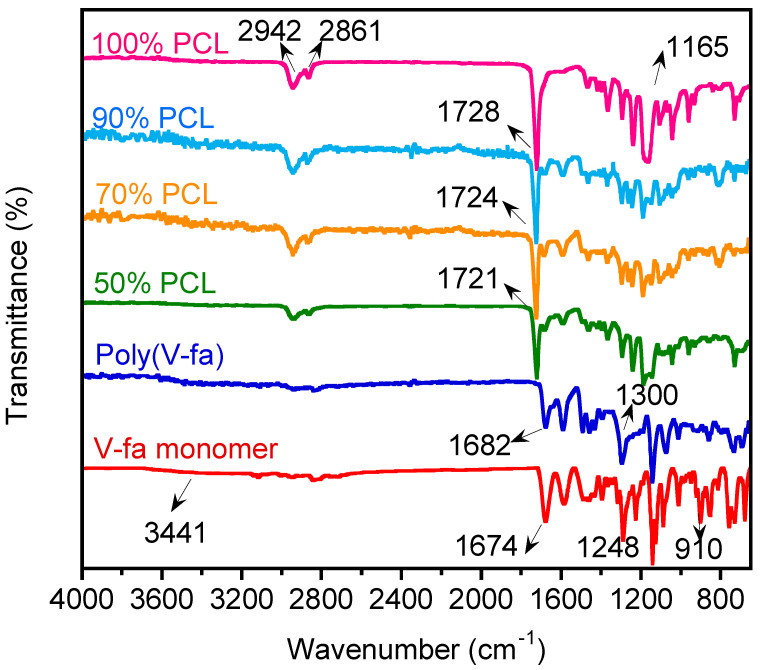
Infrared spectra of blends with varying 45 k PCL content.

**Figure 3 polymers-16-03391-f003:**
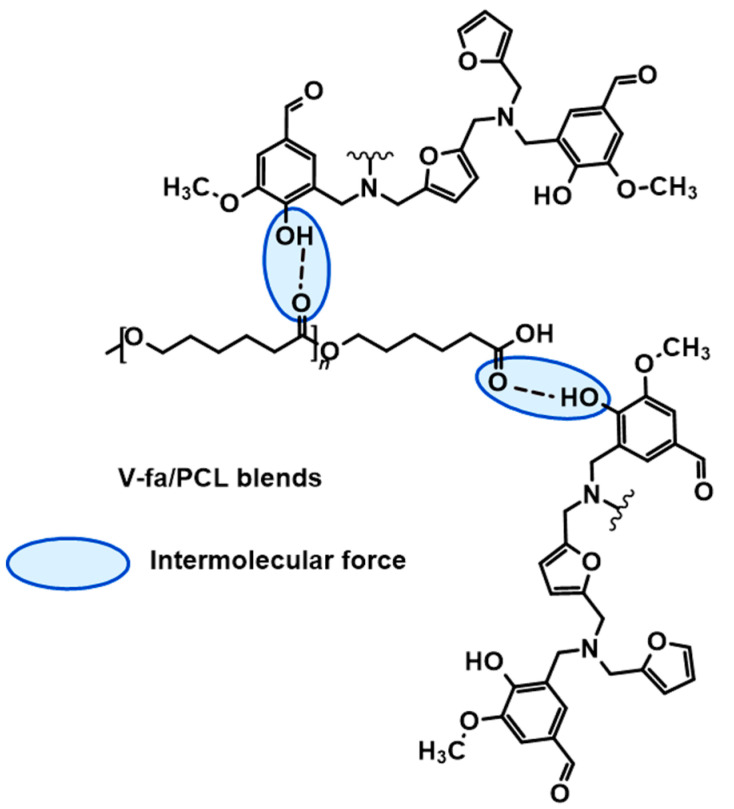
Proposed interaction between PCL and V-fa.

**Figure 4 polymers-16-03391-f004:**
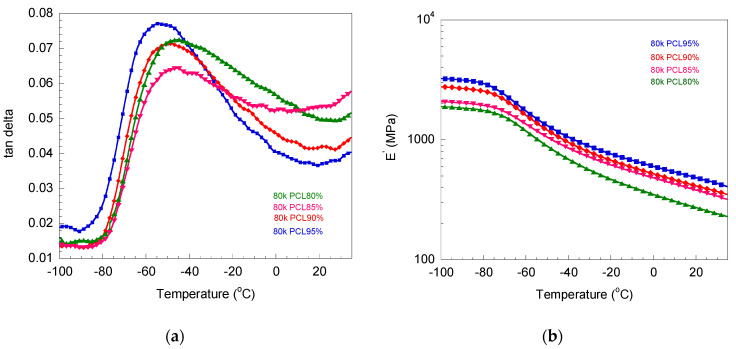
DMA thermograms of blends between V-fa polybenzoxazine and polycaprolactone with molecular weight of 80,000 Da at 80–95 wt%: (**a**) loss tangent and (**b**) storage modulus.

**Figure 5 polymers-16-03391-f005:**
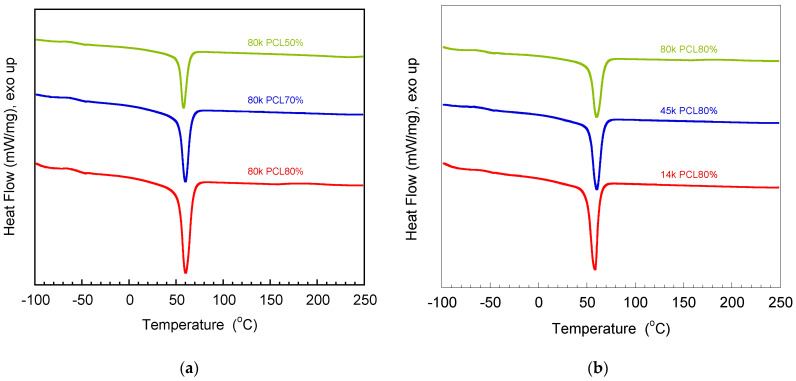
DSC thermograms of V-fa/PCL blends with (**a**) different PCL content and (**b**) different PCL molecular weights.

**Figure 6 polymers-16-03391-f006:**
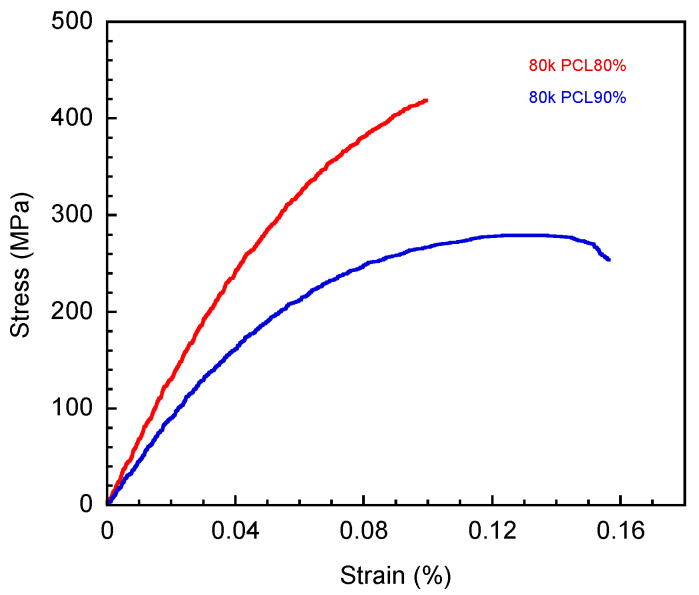
Stress–strain curve from tensile test of V-fa/80 k PCL at 80 and 90 wt%.

**Figure 7 polymers-16-03391-f007:**
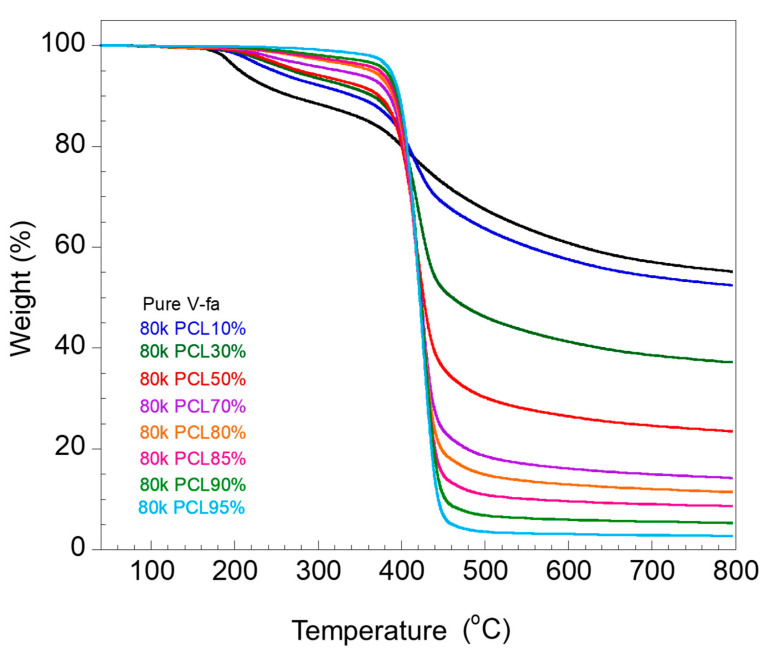
TGA thermograms of V-fa/80 k PCL blends with varying PCL content.

**Figure 8 polymers-16-03391-f008:**
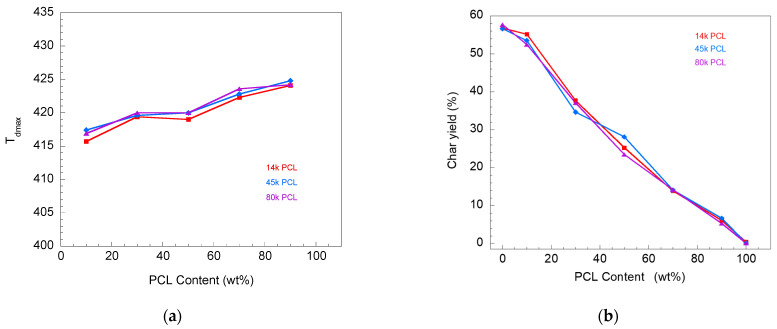
Thermal stability parameters of V-fa/PCL blends with different PCL molecular weights: (**a**) decomposition temperature at maximum decomposition rate and (**b**) char yield at 800 °C.

**Figure 9 polymers-16-03391-f009:**
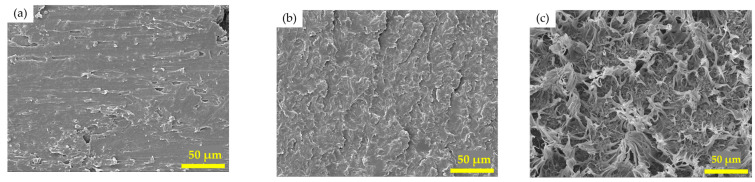
SEM images of the fracture surface for the V-fa/PCL blends at 80 wt% PCL: (**a**) 14 k PCL, (**b**) 45 k PCL, and (**c**) 80 k PCL (500× magnification).

**Figure 10 polymers-16-03391-f010:**
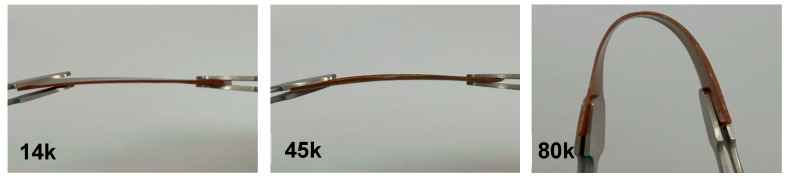
Bending test of blends with different molecular weights of PCL.

**Figure 11 polymers-16-03391-f011:**
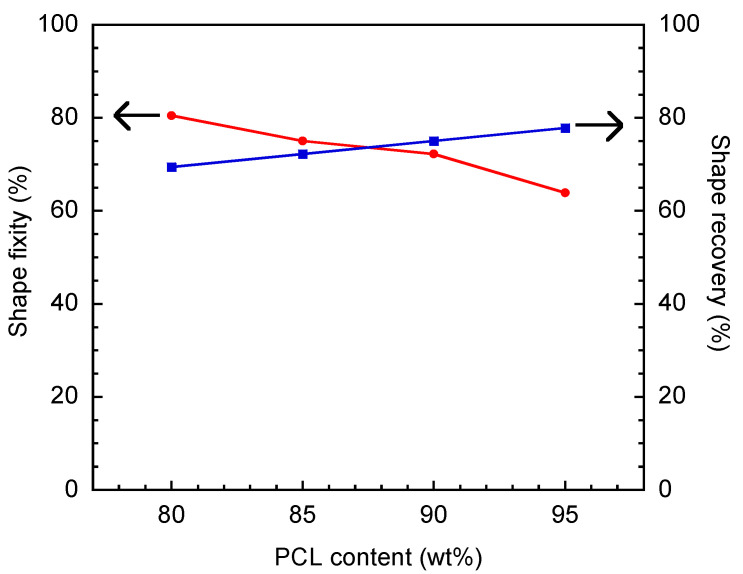
Shape fixity ratio and shape recovery ratio of V-fa/80 k PCL.

**Figure 12 polymers-16-03391-f012:**
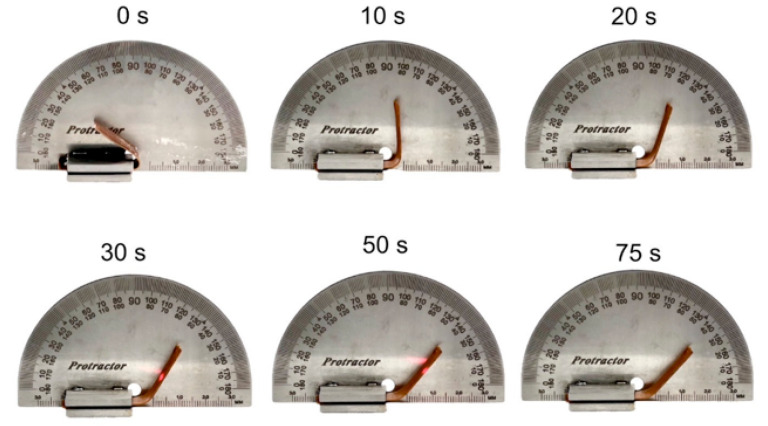
Snapshot of shape recovery process for V-fa/80 k PCL at 95 wt% PCL.

**Table 1 polymers-16-03391-t001:** Flexural modulus and flexural strength of V-fa/PCL blends.

Sample	Flexural Modulus (MPa)	Flexural Strength (MPa)
14 k PCL 90%	1629 ± 482	8.9 ± 1.9
45 k PCL 85%	1327 ± 176	16.0 ± 4.3
45 k PCL 90%	1312 ± 185	14.4 ± 1.9
45 k PCL 100%	1249 ± 266	10.7 ± 0.4
80 k PCL 90%	1005 ± 122	53.2 ± 5.1
80 k PCL 100%	870 ± 162	38.4 ± 6.3

**Table 2 polymers-16-03391-t002:** Shape recovery ratio and recovery time of V-fa/80 k PCL blends at various cycles of recovery process.

Cycle Number	Shape Recovery Ratio (%)				Recovery Time (Seconds)			
	Polycaprolactone Content (%)				Polycaprolactone Content (%)			
	80	85	90	95	80	85	90	95
1	69	72	75	78	180	175	150	75
2	64	67	67	67	185	180	160	90
3	61	61	64	67	195	180	175	160
4	61	61	64	67	220	185	180	160
5	61	61	61	67	225	185	180	170

## Data Availability

The original contributions presented in the study are included in the article, further inquiries can be directed to the corresponding author.

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
