# Peer review of "Study on the Mechanical and Thermal Properties and Shape Memory Behaviors of Blends of Bio-Based Polybenzoxazine and Polycaprolactone with Different Molecular Weights"

_polymers, 2024, doi:10.3390/polym16233391_

Round 1

Reviewer 1 Report

Comments and Suggestions for Authors

This paper describes the mechanical, thermal properties, and shape memory behaviours of the polybenzoxazine and polycaprolactone blend. While this is a reasonable effort, I don't find convincing reasons to recommend publication in Polymers. Some comments should be considered before resubmitting to Polymers or elsewhere.

 1. The monomers of benzoxazine resin (Vanilin, furfurylamine, and paraformaldehyde) and polycaprolactone were obtained from commercial sources. I do not understand how and why they called them bio-based systems (merely having furfuryl does not qualify them as bio-based).

2.     Structural characterization (such as NMR, GPC, etc.) is missing for the polymer poly(V-fa).

3.     Figure 2: Authors should present spectra in the 4000-400 cm-1 range. Also, the Y-axis is not absorbance but %T. I believe overlaid spectra would be more useful than stacked ones. Also, a panel showing the IR of the monomer V-fa and polymer poly(V-fa) should be provided and discussed.

4.     Figure 3: I do not see any reason for the proposed interaction. Why don’t authors use IR to monitor the H-bonding between the two species? How did they exclude the possibility of intermolecular H-bonding between poly(V-fa) chains

5. The structure drawn is poorly done. The authors did not consider angles, length, etc.

6. The discussion contains few references, so the authors failed to support their findings.

Comments on the Quality of English Language

In many places, authors are unclear about the content and the science they want to deliver to readers. The quality of the English is low, and it should be revised to improve the overall readability of the manuscript.

Author Response

  1. The monomers of benzoxazine resin (Vanilin, furfurylamine, and paraformaldehyde) and polycaprolactone were obtained from commercial sources. I do not understand how and why they called them bio-based systems (merely having furfuryl does not qualify them as bio-based).

Reply to reviewer

  1. Thank you for pointing this out. We agree to this comment. Therefore, we explain a definition of bio-based materials in the manuscript as below:

“The approach of bio-based materials, which are derived wholly or partly from natural resources [6], not only decrease dependence on fossil fuels but also reduce the environmental impact upon production.”

Moreover, the word of “V-fa” has been used as “bio-based polybenzoxazine” in many references such as [6-8].

            This change can be found on Page 2, Line1.

  1. Structural characterization (such as NMR, GPC, etc.) is missing for the polymer poly(V-fa).

Reply to reviewer

Thank you for the useful comment. The structure characterization using NMR has been mentioned in the article as recommended.  This change can be found in Section 2.3 and 3.1.

  1. Figure 2: Authors should present spectra in the 4000-400 cm-1 Also, the Y-axis is not absorbance but %T. I believe overlaid spectra would be more useful than stacked ones. Also, a panel showing the IR of the monomer V-fa and polymer poly(V-fa) should be provided and discussed.

Reply to reviewer

In Figure 2, the spectra have been presented in the 4000-600 cm-1 as recommended. The wavenumber was limited to 600 cm-1 (not 400 cm-1) due to the type of crystal in ATR-FTIR technique.  The Y-axis is changed to %T as suggested. The IR-spectra for the monomer V-fa and polymer poly(V-fa) have been provided as commented.

  1. Figure 3: I do not see any reason for the proposed interaction. Why don’t authors use IR to monitor the H-bonding between the two species? How did they exclude the possibility of intermolecular H-bonding between poly(V-fa) chains

Reply to reviewer

Thank you for the useful comment. The H-bonding between two polymeric components can been discussed from the shift of C=O groups as previously mentioned in previous studies [3, 4, 8, 15]

[3] Ishida, H.; Lee, Y.H. Synergism observed in polybenzoxazine and poly(ε-caprolactone) blends by dynamic mechanical and thermogravimetric analysis. Polym. 2001, 42, 6971.

[4] Ishida, H.; Lee, Y.H. Study of hydrogen bonding and thermal properties of polybenzoxazine and poly-(e-caprolactone) blends. J. Polym. Sci. Part B: Polym. Phys. 2001,39, 736

.

[8] Tiptipakorn, S.; Angkanawarangkana, C.; Rimdusit, S.; Hemvichian, K.; Lertsarawut, P. Investigation of Multiple Shape Memory Behaviors, Thermal and Physical Properties of Benzoxazine Blended with Diamino Polysiloxane. Polymers, 2023, 15(18),3814.

[15] Tiptipakorn, S.; Keungputpong, N.; Phothiphiphit, S.; Rimdusit, S. Effects of polycaprolactone molecular weights on thermal and mechanical properties of polybenzoxazine. J. Appl. Polym. Sci., 2015, 132(18), 41915.

  1. The structure drawn is poorly done. The authors did not consider angles, length, etc.

Reply to reviewer

The chemical structure has been redrawn as suggested.

  1. The discussion contains few references, so the authors failed to support their findings.

Reply to reviewer

Thank you for the useful comment. We agree to this comment. Therefore, we include the related references to support our findings as suggested.

Reviewer 2 Report

Comments and Suggestions for Authors

In the paper, authors presented the SMPs properties of PBZ/Polycaprolactone blends, and the effects of molecular weight of PCL were studied while bio-based benzoxazine monomer V-fa was used. It is good to extend the usage of bio-based benzoxazine resins.
Some suggestions are listed as follows:
1.The background of SMPs based on PBZ is lost many information which can not reveal the current progress in the SMPs of PBZ. Some of important references should be cited such as some pubilications related to blending of PBZ/Polycaprolactone around 2000~2005.

2.The information of bio-based benzoxazine monomer (V-fa) should be presented with FTIR, NMR, and Element analysis in the article or in ESI. If V-fa has been reported, authors could be make a comparision with the reported data.

3.There are many kinds of bio-based benzoxazine monomers, while V-fa is used in the paper, what is the reason using  V-fa  as BZ resin?

4. What is the differenc for this paper with reference 11?

5. In Figure 4, authors discussed the DMA thermograms of the blends between V-fa polybenzoxazine and polycaprolactone with the molecular weight of 80,000 Da at 80-95 wt%. What about the effects for polycaprolactone with different molecular weight on the DMA of the blend eg. with the same loading of 80%?

6.In the Figure 9, what are the contents of PCL of different molecular weight in the blends?

Author Response

In the paper, authors presented the SMPs properties of PBZ/Polycaprolactone blends, and the effects of molecular weight of PCL were studied while bio-based benzoxazine monomer V-fa was used. It is good to extend the usage of bio-based benzoxazine resins.
Some suggestions are listed as follows:
1.The background of SMPs based on PBZ is lost many information which can not reveal the current progress in the SMPs of PBZ. Some of important references should be cited such as some pubilications related to blending of PBZ/Polycaprolactone around 2000~2005.

Reply to reviewer

Thank you for pointing this out. We agree to this comment. Therefore, the background of SMPs based on PBZ and some publications related to PBZ/ Polycaprolactone around 2000~2005 has been included in the article as suggested.

2.The information of bio-based benzoxazine monomer (V-fa) should be presented with FTIR, NMR, and Element analysis in the article or in ESI. If V-fa has been reported, authors could make a comparison with the reported data.
Reply to reviewer

Thank you for the useful suggestion. The information of NMR has been included in the article as commented.

3.There are many kinds of bio-based benzoxazine monomers, while V-fa is used in the paper, what is the reason using V-fa as BZ resin?

Reply to reviewer Thank you for the interesting comment. Actually, there are many types of bio-based benzoxazine monomers, such as V-fa, E-fa and C-fa. The reason for selecting V-fa as BZ resin in the study is that this kind of bio-based monomer possess comparatively high cross-linked density with good mechanical properties. In comparison, this kind of monomer is more easily polymerized with low curing temperature.  This explanation has been included in the Introduction section as commented.

  1. What is the difference for this paper with reference 11?

Reply to reviewer

Thank you for your comment. This paper was related to the bio-based polybenzoxazine blended with PCL and their properties including shape memory behaviors, while the reference 11 was related to petroleum based polybenzoxazine. Moreover, the shape memory behaviors were not mentioned in reference 11 [new Ref. number of 14]. The differences have been mentioned in the introduction section.

  1. In Figure 4, authors discussed the DMA thermograms of the blends between V-fa polybenzoxazine and polycaprolactone with the molecular weight of 80,000 Da at 80-95 wt%. What about the effects for polycaprolactone with different molecular weight on the DMA of the blend eg. with the same loading of 80%?

Reply to reviewer Thank you for the interesting comment. The samples with the PCL molecular weight of 14,000 Da and 45,000 Da could not be bendable as presented in the bending test. The DMA analysis of the samples at these PCL molecular weights could not be conducted; the samples were broken. Therefore, the effects for PCL molecular weights were not presented.

6.In the Figure 9, what are the contents of PCL of different molecular weight in the blends?

Reply to reviewer: The PCL contents of 80wt% has been included in the figure caption as commented.

Reviewer 3 Report

Comments and Suggestions for Authors

After reviewing this manuscript entitled (Study on Mechanical, Thermal Properties, and Shape Memory Behaviors of the Blends between Bio-based Polybenzoxazine and Polycaprolactone with Different Molecular Weights), I found it an interesting work.

The literature survey needs some updates.

Moreover, the references are not adequate are some comments require to be replied before recommending it for publication. 

a- Some complex sentences arise through the manuscript. Shorter and more comprehensive ones will be better.

b- The novelty of this work has to be expressed in the introduction section when compared to published corresponding studies.

c- The structures of Fig. 1 and 3 cannot be read easily. They have to be plotted distinctly.

d- The discussion of Fig. 9 (SEM) has to be written in detail. The scales of this figures are not legible. Clearer ones are required.

e- The references are not recent enough. Various relevant studies published in 2024 have to be cited in this manuscript.

Comments on the Quality of English Language

The language of this work has to be revised.

Author Response

  • Some complex sentences arise through the manuscript. Shorter and more comprehensive ones will be better.

Reply to reviewer

Thank you for the useful suggestion. The manuscript has been checked for more comprehensive ones as commented.

b- The novelty of this work has to be expressed in the introduction section when compared to published corresponding studies.

Reply to reviewer Thank you for pointing this out. We agree to this comment. Therefore, the novelty of this work has been expressed in the introduction section as commented.

c- The structures of Fig. 1 and 3 cannot be read easily. They have to be plotted distinctly.

Reply to reviewer

The structures of these figures have been replotted as commented.

d- The discussion of Fig. 9 (SEM) has to be written in detail. The scales of this figures are not legible. Reply to reviewer

Thank you for your comment. The scale of Figure 9a-9c has been rewritten for more clearness. More discussions in detail have been written as recommended.

e- The references are not recent enough. Various relevant studies published in 2024 have to be cited in this manuscript.

Reply to reviewer

Thank you for your comment. The relevant studies published in 2024 were added in the manuscript as suggested.

Reviewer 4 Report

Comments and Suggestions for Authors

This article is interesting and nicely written. However, the following revisions are required before accepting this manuscript.

Comments:

1. In the FTIR graph, important peaks should be shown at the arrow mark.

2.  More discussion with relevant refs is needed in the DSC, DMA sections.

3. TGA part is also more discussion is needed.

4. What about biocompatibility of the prepared shape memory blends.

Comments on the Quality of English Language

Minor English language corrections are needed

Author Response

  1. In the FTIR graph, important peaks should be shown at the arrow mark.

Reply to reviewer: Thank you for pointing this out. We agree to this comment. The important peaks in FTIR graph have been shown at the arrow mark as suggested.

  1. More discussion with relevant refs is needed in the DSC, DMA sections.

Reply to reviewer

More discussion with the related references has been included in the DSC and DMA sections as comments.

  1. TGA part is also more discussion is needed.

Reply to reviewer

More discussion has been mentioned in the TGA section as suggested.

  1. What about biocompatibility of the prepared shape memory blends.

Reply to reviewer

Thank you for your useful comment. The biocompatibility of the prepared shape memory blends is the crucial characteristics to be further studied and not included in the manuscript.

Round 2

Reviewer 1 Report

Comments and Suggestions for Authors

The authors have revised their manuscript satisfactorily. I have just two additional suggestions:

1. Fig. 1, please provide the NMR data side by side, as the scales do not match the values.

2. Please check and correct the abbreviations (such as PCL in Table 2) throughout the manuscript. 

Reviewer 2 Report

Comments and Suggestions for Authors

Authors made improvements according to the comments suggested by reviewers in this revised version.

Some suggestions are as follows,

1. Although authors the V-fa and poly(V-fa) characterized by Solid 13C NMR, as for as readers concern, V-fa has been reported in reference 33, therefore the V-fa should be presented with H-NMR in the section of material in the present paper , or it could be presented detail in FTIR, NMR, and Element analysis in Support Information.
From H-NMR, then readers could learn that the purity of V-fa compared with that of reference 33.

2.The format of reference 7 is incorrect.

Reviewer 3 Report

Comments and Suggestions for Authors

The revised form of the manuscript (Study on Mechanical, Thermal Properties, and Shape Memory Behaviors of the Blends between Bio-based Polybenzoxazine and Polycaprolactone with Different Molecular Weights) seems better than before.

The authors have responded properly to my remarks.

I can recommend it to be accepted for publication in the current form.